# Synthesis of a Novel Bifunctional Epoxy Double-Decker Silsesquioxane: Improvement of the Thermal Stability and Dielectric Properties of Polybenzoxazine

**DOI:** 10.3390/polym14235154

**Published:** 2022-11-26

**Authors:** Xiaoyi Sun, Jing Wang, Qixuan Fu, Qian Zhang, Riwei Xu

**Affiliations:** 1Key Laboratory of Carbon Fiber and Functional Polymers, Beijing University of Chemical Technology, Ministry of Education, Beijing 100029, China; 2Beijing Information Technology College, Beijing 100029, China; 3Ablestik (Shanghai) Co., Ltd., No. 332, Meigui South Road, WGQ Free Trade Zone, Shanghai 200131, China

**Keywords:** double-decker silsesquioxane, epoxy group, polybenzoxazines, thermal stability, dielectric properties

## Abstract

In this study a new type of bifunctional epoxy compound (DDSQ-EP) based on double-decker silsesquioxane (DDSQ) was synthesized by process of alkaline hydrolysis condensation of phenyltrimethoxysilane and corner capping reaction with dichloromethylvinylsilane, followed by epoxidation reaction of vinyl groups. The resultant structures were confirmed using Fourier transform infrared spectrometry, nuclear magnetic resonance spectrometry and time-of-flight mass spectrometry, respectively. The DDSQ-EP was incorporated into polybenzoxazine to obtain the PBZ/DDSQ-EP nanocomposites. The uniform dispersion of DDSQ-EP in the nanocomposites was verified by X-ray diffraction and scanning electron microscope. The reactions occurred during the curing of the composites and were investigated using infrared spectroscopy of segmented cures. Dynamic mechanical analysis and thermal gravimetric analysis indicated that the storage modulus, glass transition temperature and thermal stability of PBZ/DDSQ-EP were increased in comparison with pure benzoxazine resins. Assessment of dielectric properties demonstrated that the dielectric permittivity and dielectric loss of polybenzoxazine decreased slightly because of the addition of DDSQ-EP.

## 1. Introduction

Polybenzoxazine (PBZ) consists of phenolic-like materials with a similar precursor and chemical structure to phenolic resins [1]. Compared with other types of traditional thermosets (e.g., epoxy resins, phenolic resins), PBZ thermosets possess excellent mechanical properties, dimensional stability, thermal stability and flame retardance [2]. Another attractive feature of PBZ is that they can be obtained by thermoactivated ring-opening polymerization of benzoxazine monomers without a catalyst (Figure 1), no small molecules are generated during the curing process, and the finished product has low porosity [3,4]. These advantages have attracted the attention of researchers. The molecular structure design of PBZ is very flexible, and various structures of polybenzoxazines have been developed for a wide range of applications, including polymer coatings [5,6,7], flame retardants [8,9,10], electromagnetic shielding [11,12], and microelectronics [1,13,14,15]. Nevertheless, PBZ thermosets also have some drawbacks that limit its application. For instance, the glass transition temperature (Tg) of unmodified bisphenol-A/aniline polybenzoxazine (PB-a) is generally 170–200 °C, which is difficult to use as a matrix for high-performance composites [16,17,18,19]. Furthermore, the crosslinking densities of PBZ are low, and the mechanical strength of the thermosets in the rubber state is quite poor [3,20]. In addition, polybenzoxazine thermosets can be used as a microelectronic packaging material, further reducing the dielectric constant, and loss of benzoxazine resin is also the target of the researchers. Some reports have focused on mixing various inorganic compounds (e.g., clay [21,22], graphene [23,24,25], POSS [26,27]) with benzoxazine monomers to form organic–inorganic nanocomposites to improve the crosslinking density, the thermal stability of thermosets and reduce the dielectric constant.

Polyhedral oligomeric silsesquioxane (POSS) is a cage-type organic/inorganic hybrid structure with organic groups of various designs for copolymerization, blending or grafting [28,29,30,31,32]. The polymer/POSS nanocomposites formed by mixing POSS into thermosets have a significant impact on thermal stability, mechanical properties, dielectric properties and flame retardance of the resin, and their thermal stability and mechanical strength are better than pure resin [33]. The hollow structure of cage-type makes it an ideal low-dielectric nanometer filler [34,35,36,37]. In the past few years, there has been ample literature reported on the use of POSS to modify PBZ. However, simply adding POSS to the resin by physical blending typically causes the problem of aggregation of POSS particles. Additionally, the crosslinked network of the resin will be destroyed and the mechanical properties will be degraded. The compatibility and reactivity between POSS and the resin matrix are effectively increased if the organofunctional groups on the vertices of the cage structure are designed [38,39]. In our previous study, in order to improve the compatibility of benzoxazine with POSS, bisoxazoline was used as a compatibilizer for octaaminophenyl POSS and PBZ [40], benzoxazine-functionalized POSS was also synthesized [40] and the properties of benzoxazine resin were successfully improved. In general, the structure of polymer/POSS nanocomposites depends on the functional structure of POSS nanoparticles. In most cases, the monofunctionalized POSS is located at the polymer chain end or side chain and multifunctional POSS nanoparticles added to the polymer matrix typically produces insoluble cross-linked polymers. It is worth nothing that the addition of a low content of bifunctional POSS to the polymer can generally form polymer/POSS nanocomposites with the POSS unit in the main chain [41,42,43,44,45]. The use of bifunctional POSS for thermoset plastic provides new approaches for the development of novel polymer/POSS nanocomposites. Liao et al. synthesized a bifunctional phenolic compound and then synthesized bis-allyl benzoxazine double-decker silsesquioxane derivative by Mannich condensation to obtain highly thermally stable and transparent benzoxazine nanocomposites [46]. Niu et al. synthesized 3,13-diglycidyloxypropyloctaphenyl double-decker silsesquioxane by three-step method and modified the phenolic resin, as adding a small amount of POSS can improve the heat resistance of the system without affecting the curing temperature of the PF resin itself [47]. Zhang et al. combined this bifunctional epoxy bilayer sesquisiloxane with benzoxazine monomers, causing the epoxy group on the POSS to react with the phenolic hydroxyl group on PBZ to form additional crosslinking. Additionally, POSS cage has nano-strengthening effect, resulting in the Tg of the nanocomposite material is significantly improved in stark contrast to pure PBZ [48]. The epoxy group is directly connected to the silicon atoms on the POSS cage and is expected to achieve better thermodynamics, thermal stability and tensile properties, and lower dielectric properties. To the best of our knowledge, the existing literature on the preparation of bifunctional epoxy compound based on double-decker silsesquioxane is to use hydrosilylation method to connect the epoxy group and the POSS cage structure through the carbon chain. There are no reports on the modification of benzoxazines thermosets by synthesizing nanomaterials in which epoxy groups are directly connected to silicon atoms on POSS cages.

In this study, we prepared and characterized a novel bifunctional epoxy double-decker silsesquioxane (DDSQ-EP) and then PBZ/DDSQ-EP nanocomposites were prepared by simple solution blending. Compared with other bifunctional epoxy POSS, DDSQ-EP can be synthesized by a simple method. Due to the reactivity of polybenzoxazine with the phenolic hydroxyl groups produced when the epoxy groups were cured, DDSQ-EP can participate in the crosslinking of PBZs, thereby adding additional crosslinking points to achieve chemical modification of polybenzoxazines. Finally, we prepared and studied the structure, morphology, thermodynamic properties and dielectric properties of BZ/DDSQ-EP nanocomposites.

## 2. Materials and Methods

### 2.1. Materials

Bisphenol A-based benzoxazine (BZ) monomer was synthesized from aniline, bisphenol A and paraformaldehyde by using a solvent method described in our previous paper [49]. Phenyltrimethoxysilane (PTMS) was supplied by Zhejiang Chemical Technology Company, Hangzhou, China. Dichloromethylvinylsilane (DMVS) and triethylamine were supplied by Alfa-Aesar, Tianjin, China. Sodium hydroxide (NaOH), isopropanol, toluene, acetone, chloroform, hydrogen peroxide(H_2_O_2_), acetic acid (HAc) and concentrated sulfuric acid (H_2_SO_4_) were purchased from Beijing Chemical Works, Beijing, China.

### 2.2. Synthesis of DDSQ-VS

The DDSQ-VS was synthesized according to the methods outlined in reference [50]. It was a process of alkaline hydrolysis condensation of PTMS and then corner capping reaction with DMVS. Firstly, the double-decker silsesquioxane tetrasodium salt (DD-Na) was synthesized by silane hydrolysis with PTMS and sodium hydroxide in a yield of up to 70%. The specific method is as follows: exactly 35.7 g (0.18 mol) PTMS, 4.8 g (0.12 mol) NaOH, 3.6 g (0.2 mol) H_2_O, and 200 mL isopropanol were added in a 500 mL round-bottomed flask. The mixture was heated and kept under reflux for 4 h. Subsequently, the system was cooled to room temperature and the reaction was conducted for another 15 h. Afterward, the solvent was removed by filtration. The residue was washed with isopropanol three times and dried in a vacuum oven at 40 °C for 24 h to obtain DD-Na. Secondly, the silylation reaction between DD-Na and DMVS was carried out. Exactly, 5.8 g (5 mmol) DD-Na, 2.8 mL (20 mmol) triethylamine, and anhydrous toluene 50 mL were added in a 250 mL round-bottomed flask with a magnetic stirrer and vigorous stirring. The flask was immersed in an ice-water bath and then purged with highly pure nitrogen for one hour. Thereafter, 1.4 mL (10 mmol) DMVS dissolved in 10 mL of anhydrous toluene were added dropwise within 30 min. The reaction was performed at −5 °C for 2 h and at room temperature for 10 h. The insoluble solids (i.e., sodium chloride) were removed by filtration and the solvents together with other volatile compounds were eliminated via rotary evaporation to afford the white solids. The solids were washed with methanol three times and dried in vacuo at 40 °C for 24 h; the product was obtained with a yield of 53%.

### 2.3. Synthesis of DDSQ-EP

The DDSQ-VS was epoxidated under the condition of catalysis. Exactly, 6.02 g (5 mmol) DDSQ-VS dissolved in 30 mL chloroform was added into a 100 mL flask. Subsequently, 0.6 mL (10 mmol) HAc and 0.2 mL H_2_SO_4_ were added under nitrogen protection. The mixture was stirred with a magnetic stirrer and heated up to 70 °C. Then 3 mL (10 mmol) H_2_O_2_ was slowly dropped into the flask and kept under reflux for 6 h. The reaction solution was washed successively with sodium carbonate solution and deionized water several times, and the aqueous phase was removed by the separation funnel. Then the organic phase was distilled via rotary evaporation to afford the white solids. This method was simple, effective and environmentally friendly, and the yield is up to 60%.

### 2.4. Preparation of PBZ/DDSQ-EP Nanocomposites

DDSQ-EP and benzoxazine monomers were dissolved in acetone at a ratio of 100:0, 99.5:0.5, 99:1, 98:2, 96:4 to obtain PBZ/DDSQ-EP mixture. The mixing was dispersed for 30 min by ultrasonic, and the acetone solvent was volatilized at room temperature, then placed in a vacuum oven for 12 h under 45 °C to dry thoroughly. Then, the mixture was cured according to procedure with 140 °C/2 h + 160 °C/2 h + 180 °C/2 h + 200 °C/2 h + 220 °C/2 h. After curing was completed, the pure PBZ resin and 0.5 wt.%, 1 wt.%, 2 wt.% and 4 wt.% PBZ/DDSQ-EP nanocomposites are obtained.

### 2.5. Characterization

Fourier transform infrared spectrometry (FTIR) of the samples were examined by Nicolet-670; the wavenumber range was 4000 cm^−1^ to 400 cm^−1^. DDSQ-EP was mixed with KBr and pressed into sheets for testing. The samples were dissolved in CDCl_3_, the ^1^H nuclear magnetic resonance (NMR) spectra and ^29^Si NMR spectra of the samples were conducted by Bruker Avance III 400 MHz at 25 °C. The Matrix-assisted laser desorption ionization time-of-flight mass spectrometry (MALDI-TOF) was conducted by BIFLEX III (Bruker Daltonics Inc., Billerica, MA, USA). Samples were dissolved in THF, and CHCA was used as matrix. Dynamic mechanical analysis (DMA) was determined by Rheometric Scientific DMTA V with a heating rate of 3 °C/min at 1 Hz, and the test temperature range was from 50 °C to 300 °C. Scanning electron microscope (SEM) images were recorded using a Hitachi S-4700 operated at 5 kV. Thermal gravimetric analysis (TGA) was determined by TA-Q50 from 25 °C to 800 °C with a heating rate of 10 °C/min under nitrogen atmosphere. The X-ray diffraction (XRD) spectrum of the samples was conducted by D/MAX 2500 VBZ + /PC with Cu-K-α radiation (λ = 0.154 nm) under scanning speed 5°/min, and the scanning range was from 3° to 50°. The dielectric constant and dielectric loss of the nanocomposites were measured by dielectric constant tester (WY2851-Q) over a frequency range from 40 Hz to 30 MHz at ambient temperatures. All the 10.0 mm diameter cylindrical samples were polished with thicknesses of 1 mm. The dielectric constant and dielectric loss of each sample were measured at least three times, and the final results were the average of three measurements.

## 3. Results and Discussion

### 3.1. Synthesis of DDSQ-VS and DDSQ-EP

Figure 2 presents the preparation of the DDDOVS and DDSQ-EP. Each intermediate chemical structure can be confirmed by FTIR, ^1^H NMR, ^29^Si NMR and MALDI-TOF analyses. Figure 3 demonstrates the FTIR spectrum of DD-Na, DDSQ-VS and DDSQ-EP. It can be seen that the above three different materials have characteristic absorption peaks of Si-O-Si at 1132 cm^−1^ and there is no change before and after the reaction, which indicates that the cage structure of POSS is complete. Compared with DD-Na, the new peak of DDSQ-VS at 1409 cm^−1^ is the characteristic absorption peak of Si-CH=CH_2_ and the characteristic peak of stretching vibration of Si-CH_3_ appears at 809 cm^−1^ and 774 cm^−1^, indicating that DDSQ-VS was successfully synthesized. In the DDSQ-EP spectrum, the characteristic absorption peak of the vinyl group at 1409 cm^−1^ disappears, while the asymmetric stretching vibration peak of C-O-C at 873 cm^−1^ and the symmetric stretching vibration peak at 1232 cm^−1^ appears which demonstrates that the vinyl group has been completely oxidized to the epoxy group.

The ^1^H NMR spectra of DDSQ-VS and DDSQ-EP are demonstrated in Figure 4. For DDSQ-VS, the signals of resonance assignable to the proton hydrogen on the phenyl ring and vinyl were detected at 7.14~7.58 ppm and 6.0~6.19 ppm, respectively. It is a proton hydrogen on the methyl group attached to Si at 0.41 ppm. The ratio of integral intensity for these three signals of resonance was measured to be 20:3:3, which was exactly identical with theoretical value. For DDSQ-EP, the vinyl proton peak disappears at a chemical shift of 6.0 to 6.19 ppm, indicating complete epoxidation of the double bond. The peaks of resonance at 6.0~6.19 ppm, 2.38~2.87 ppm, 7.20~7.57 ppm correspond to the proton hydrogen on the epoxy group, benzene ring and silicon methyl.

Demonstrated in Figure 5 are the ^29^Si NMR spectra of DDSQ-VS and DDSQ-EP. As it can be seen that both of them have three main signal peaks. For DDSQ-VS, the chemical shift at −31.45 ppm is the silicon atom connecting the vinyl, methyl and cage skeletons. After the epoxidation of the vinyl group, the chemical shift of the silicon atom changes to a higher position, correspondingly at −27.76 ppm in the DDSQ-EP spectrum whereas the peaks at −78.42 ppm and −79.58 ppm are assignable to the chemical shift of silicon atoms in the POSS skeleton, the position before and after the reaction is basically unchanged.

MALDI-TOF mass spectroscopy and their mass spectra are presented in Figure 6. For DDSQ-VS, the molecular weight was measured to be M = 1227.2 corresponding to the strong signal peak at 1227 m/z, which was exactly identical with theoretical value. For DDSQ-EP, the position of the single peak appears at 1259.8 m/z, indicating that the relative molecular mass is 1259.8 which is consistent with the theoretical value (1237.8 + 23). The target product was successfully confirmed by infrared spectrum, nuclear magnetic resonance spectrum, nuclear magnetic silicon spectrum and MALDI-TOF spectrum, and the purity was high.

### 3.2. Thermal Curing Properties of PBZ/DDSQ-EP

To understand the thermal polymerization mechanism of PBZ/DODDES, we use FTIR to investigate the reactions that occur during the curing of the composites. Figure 7 displays the segmented curing infrared spectrum of the PBZ/DODDES composite system. It can be seen that with the increase in temperature, the characteristic absorption peak of epoxy group at 914 cm^−1^, the characteristic peak of oxazine ring at 947 cm^−1^ and the asymmetric stretching vibration characteristic peak of Ar-O-C at 1233 cm^−1^ gradually disappear. While the trisubstituted transition on the benzene ring at 1500 cm^−1^ become tetrasubstituted, and the characteristic absorption peak of the hydroxyl group appears at 3400 cm^−1^, mainly including the phenolic hydroxyl group on polybenzoxazine and the hydroxyl group generated after ring opening of the epoxy group. It is worth noting that the contribution of hydroxyl generated after ring opening of the epoxy group is very small. The above results indicate that as the temperature increases, PBZ and DDSQ-EP via the reaction of phenolic hydroxyl groups of PBZ with epoxy groups of the difunctional POSS, form a crosslinked network structure (see Figure 8).

Figure 9 displays the plots of tanδ and storage modulus as functions of temperature for PBZ resins with various DDSQ-EP content. As it can be seen, the raise of DDSQ-EP content can increase the storage modulus and glass transition temperature of PBZ. From the overall effect, DDSQ-EP can improve the storage modulus of PB-a, and with the increase in DDSQ-EP content, the storage modulus first increases and then decreases. When the content of DDSQ-EP is 1 wt.% by weight, the storage modulus of the material is maximized. The increase and decrease in storage modulus are mainly due to the combined effect of three aspects. First, DDSQ-EP cage structure has nano strengthening effect, and the PBZ macro-molecular chain movement is difficult, which increases the rigidity of PBZ and leads to an increase in the modulus of the material. Second, the existence of hollow POSS leads to the decrease in material density and storage modulus. Finally, DDSQ-EP has a huge volume resulting in the PBZ molecular chains not being tightly arranged, and the material density and modulus decreasing. It is worth noting that the storage modulus of nanocomposites is also affected by the purity of resins, and by the curing process or the post curing process etc. The results of our analysis demonstrate that when the DDSQ-EP content is too low, the increase and decrease in storage modulus counteract each other, and the storage modulus enhancement effect is not very obvious. When the DDSQ-EP content is greater than 1 wt.%, the agglomeration phenomenon causes the nanoparticles to be dispersed unevenly which does not achieve the desired enhancement effect. The increase in the glass transition temperature (Tg) and the decrease in the loss factor indicate that the heat resistance of the material is improved and the rigidity is enhanced. This is because the addition of DDSQ-EP promotes the further crosslinking of the PBZ to form a network, and the cage structure enhances the rigidity of the resin.

TGA analyses are used to study the thermal stability of pure DDSQ-EP and the organic–inorganic nanocomposites with different DDSQ-EP content. The thermogravimetric data of DDSQ-EP and PBZ curing spline with different DDSQ-EP content are demonstrated in Figure 10. The addition of DDSQ-EP can improve the thermal stability of benzoxazine resin. With the increase of DDSQ-EP content, the 5% weight loss temperature and maximum temperature loss rate temperature rise. When the content is greater than 1 wt.%, the 5% weight loss rate temperature starts to drop. The reason may be that too much DDSQ-EP is added and the nano particles agglomerate, resulting in a decrease in the crosslinking density of the system. The benzeneoxazine resin containing DDSQ-EP has a higher carbon residual rate at 650 °C than that of pure benzoxazine resin. The overall trend is to increase first and then decrease, and the rate of carbon residue with a content of 2 wt.% is increased by 14%. Table 1 displays the TGA results of PBZ/DDSQ-EP materials in different DDSQ-EP contents.

### 3.3. Morphology and Dielectric Constants of BZ/DDSQ-EP

XRD was used to study the crystalline morphology of DDSQ-EP and PBZ with five different DDSQ-EP contents. Figure 11 displays that DDSQ-EP is a crystalline compound and PBZ is an amorphous polymer. The material obtained by adding DDSQ-EP to the BZ monomer in different percentages and solidification is also an amorphous polymer. Moreover, like the pure PBZ resin, the composite systems of DDSQ-EP with different contents have a distinct dispersion peak at 2θ ≈ 18°, demonstrating that the dispersion of these systems are in high performance and no obvious aggregation state occurs.

The morphology of the organic–inorganic thermosets were examined by means of SEM. Figure 12 exhibits the SEM of the PBZ/DDSQ-EP composite cross section. Figure 12a demonstates the first group with DDSQ-EP content of 1 wt.% and Figure 12c demonstrates the second group with DDSQ-EP content of 2 wt.%. The large dark substrate in the SEM is PBZ resin, and DDSQ-EP is not easily recognized. The distribution of silicon element in the composite was further investigated by SEM. It can be observed from Si mapping that the dispersion of DDSQ-EP is relatively uniform, manifesting that DDSQ-EP is well-distributed in the resin system (Figure 12b,d).

The dielectric constant (ε′) and dielectric loss (ε″) of different percentages of PBZ/DDSQ-EP composites are demonstrated in Figure 13. As the DDSQ-EP content increases, the permittivity property of the composite decreases. Dielectric constant and loss of PBZ/DDSQ-EP were significantly reduced through introduction of DDSQ-EP owing to their inherent hollow cores, and dielectric constant of the organic–inorganic nanocomposites were further decreased with an additional increase in DDSQ-EP. A small amount of DDSQ-EP can even reduce the dielectric property of the material. When 4 wt.% of DDSQ-EP was added, the dielectric permittivity of the material decreased from 3.75 to 3.45, and the dielectric loss tangent decreased from 0.0078 to 0.0070. 

## 4. Conclusions

A bifunctional epoxy compound that connects epoxy group directly to silicon atoms of double-decker silsesquioxane was successfully synthesized and was used to improve the performance of benzoxazine resins. DDSQ-EP was effective in reducing the dielectric constant and loss of the benzoxazine resins due to its hollow structure, moreover, it increased the storage modulus, glass transition temperature and thermal stability of benzoxazine resins. FTIR of the segmented cured product confirmed that the epoxy groups of DDSQ-EP reacted with the benzoxazine in the curing temperature range, thus additional cross-linking points were increased. When the content of DDSQ-EP is above 1 wt.%, the storage modulus and glass transition temperature of the nanocomposites are higher than those of pure benzoxazine resin, and the carbon residue rate is higher at 650 °C. The carbon residue rate of benzoxazine resin containing 2 wt.% DDSQ-EP increased by 14%. The addition of DDSQ-EP makes the dielectric permittivity and dielectric loss of the nanocomposites decrease, and the addition of 4 wt.% DDSQ-EP can obtain benzoxazine resin with the lowest dielectric permittivity of 3.45. The good compatibility between DDSQ-EP and benzoxazine resin is revealed from XRD and the Si element distribution map of SEM.

## Figures and Tables

**Figure 1 polymers-14-05154-f001:**
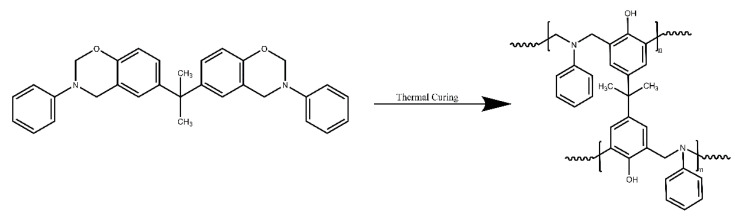
Thermally induced ring-opening polymerization of benzoxazine.

**Figure 2 polymers-14-05154-f002:**
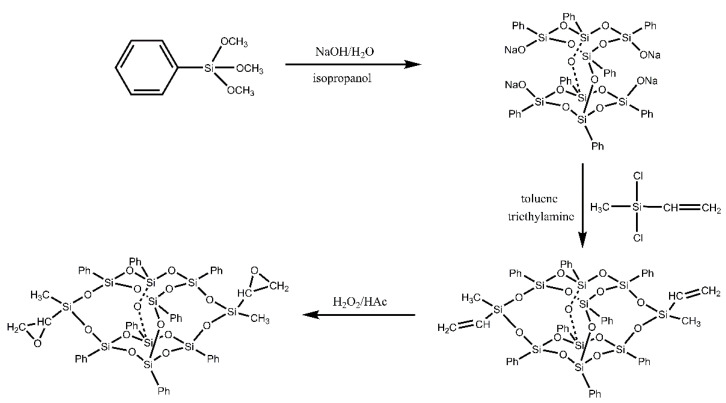
Synthetic route of DDSQ-EP.

**Figure 3 polymers-14-05154-f003:**
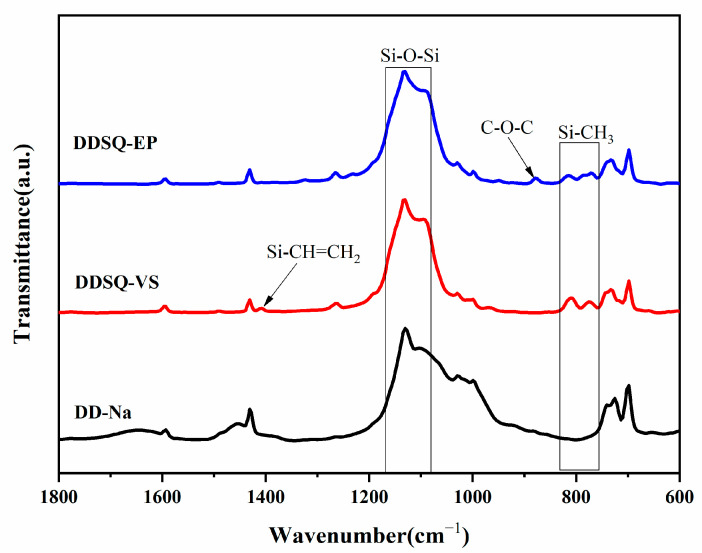
FTIR spectra of DD-Na, DDSQ-VS and DDSQ-EP.

**Figure 4 polymers-14-05154-f004:**
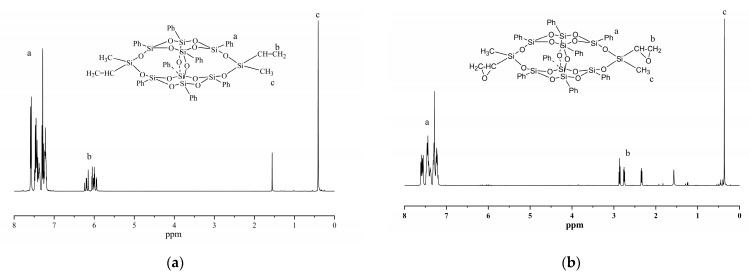
^1^H-NMR spectra of (**a**) DDSQ-VS and (**b**) DDSQ-EP.

**Figure 5 polymers-14-05154-f005:**
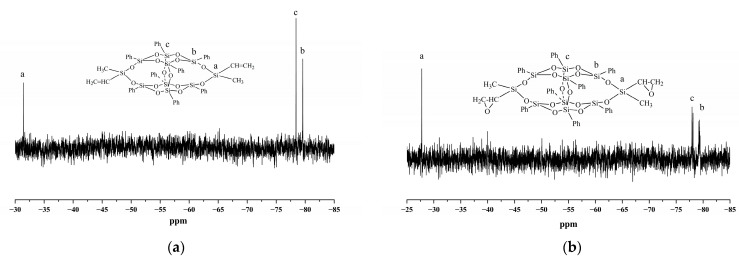
^29^Si-NMR spectra of (**a**) DDSQ-VS and (**b**) DDSQ-EP.

**Figure 6 polymers-14-05154-f006:**
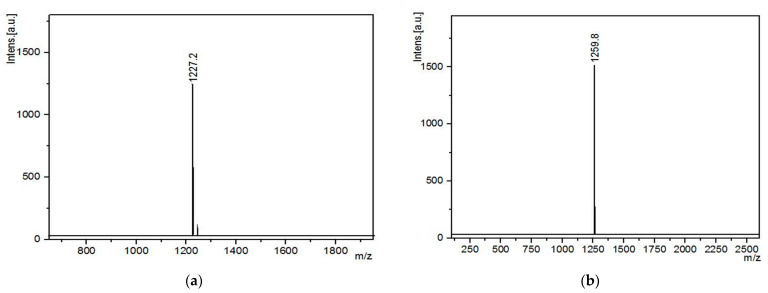
MALDI-TOF mass spectra of (**a**) DDSQ-VS and (**b**) DDSQ-EP.

**Figure 7 polymers-14-05154-f007:**
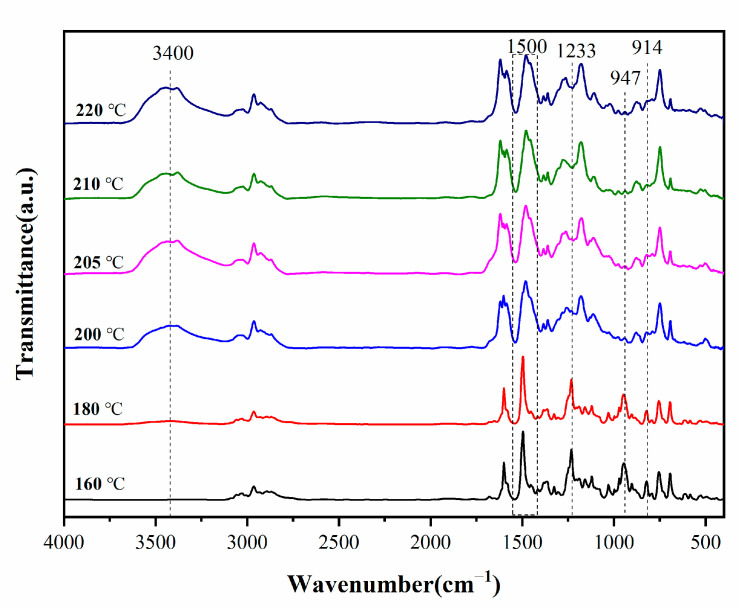
FTIR spectra of PBZ/ DDSQ-EP (2 wt.%) resins at different temperature.

**Figure 8 polymers-14-05154-f008:**
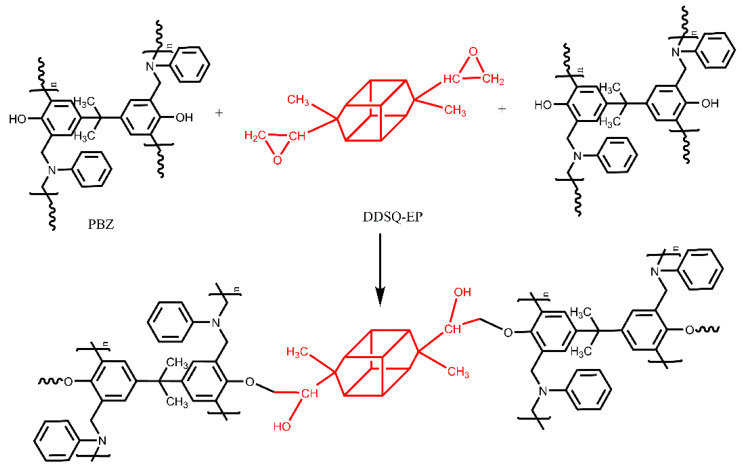
The crosslinking between PBZ and DDSQ-EP.

**Figure 9 polymers-14-05154-f009:**
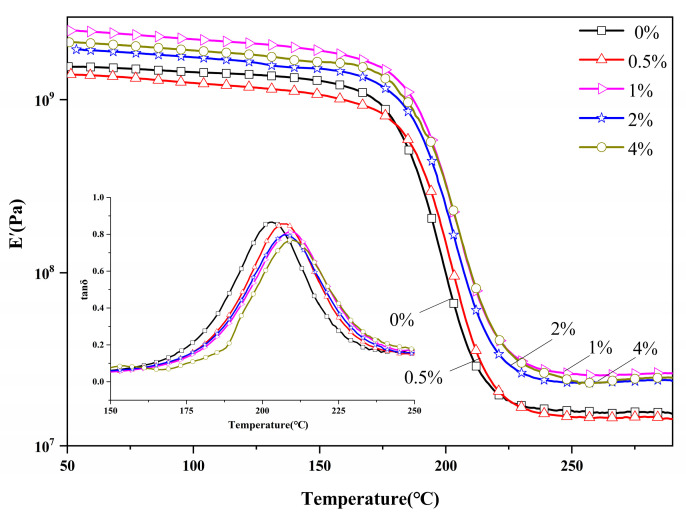
DMA curves of PBZ resins with different DDSQ-EP content.

**Figure 10 polymers-14-05154-f010:**
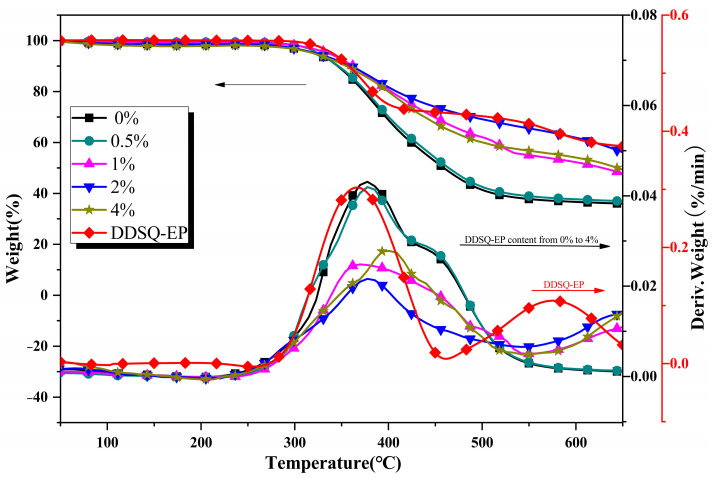
TGA curves of DDSQ-EP and PBZ resins with different DDSQ-EP content.

**Figure 11 polymers-14-05154-f011:**
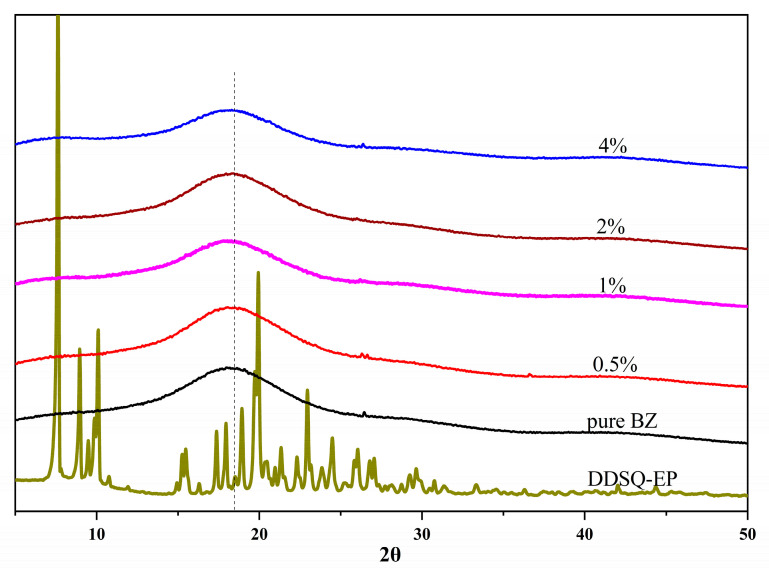
XRD curves of BZ/DDSQ-EP mixture.

**Figure 12 polymers-14-05154-f012:**
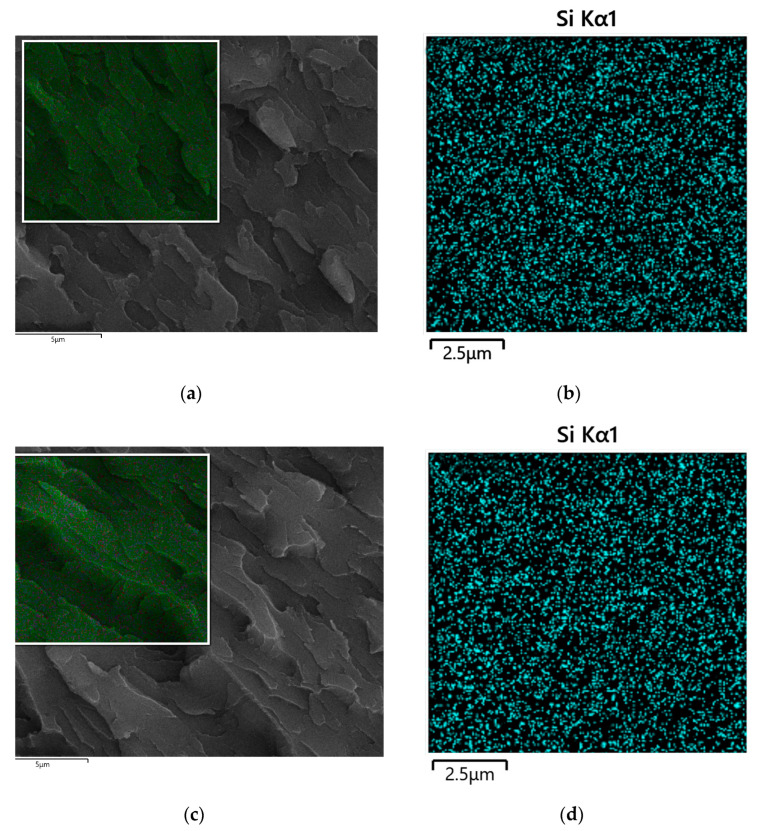
SEM and Si-mapping of BZ/DDSQ-EP composites: (**a**) The SEM of BZ/DDSQ-EP (1 wt.%), (**b**) The Si mapping of BZ/DDSQ-EP (1 wt.%), (**c**) The SEM of BZ/DDSQ-EP (2 wt.%), (**d**) The Si-mapping of BZ/DDSQ-EP (2 wt.%).

**Figure 13 polymers-14-05154-f013:**
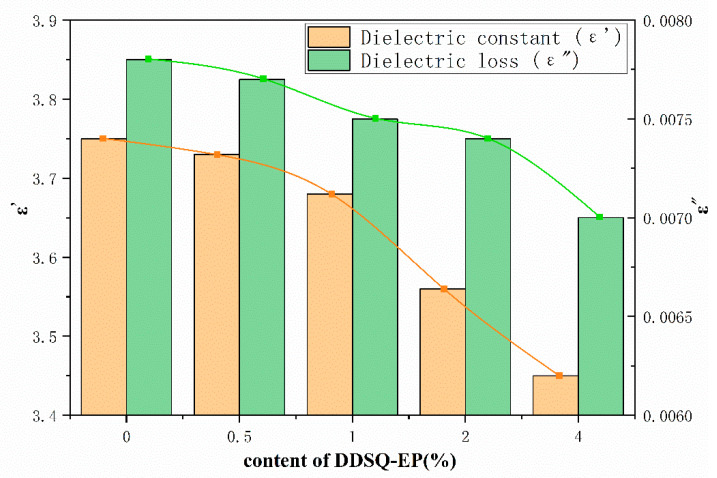
Dielectric constant (ε′) and dielectric loss (ε″) results of PBZ/DDSQ-EP with different DDSQ-EP content.

**Table 1 polymers-14-05154-t001:** TGA results of PBZ/DDSQ-EP materials in different DDSQ-EP contents.

DDSQ-EP Content (%)	0	0.5	1	2	4
Residual amount (%)	36	37	48	56	50
5% thermogravimetric temperature (°C)	314	322	330	320	314
Maximum weight loss rate temperature (°C)	337	337	362	377	393

## Data Availability

Data presented in this study are available on request from the corresponding author.

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
