# Peer review of "Synthesis of a Novel Bifunctional Epoxy Double-Decker Silsesquioxane: Improvement of the Thermal Stability and Dielectric Properties of Polybenzoxazine"

_polymers, 2022, doi:10.3390/polym14235154_

Round 1
Reviewer 1 Report
Page 1, line 32
Polybenzoxazine being a phenolic structure, it is not strong under UV irradiation. Usually, a phenolic compound is used as a UV stabilizer because it sacrifices itself to protect other polymers. Polybenzoxazine would do the same. How can polybenzoxazine be good for weather resistance then?
Page 1, line 41
“For instance, the glass transition temperature (Tg) of PBZ is only about 150°C, which is difficult to use as a matrix for 41 high-performance composites [16-19].” This statement is totally erroneous. It would be rather difficult to find polybenzoxazine with Tg 150oC among hundreds of polybenzoxazines reported in the literature. Many polybenzoxazines show Tg above 200 oC and multiple examples above 400oC. Where did you obtain such statements? The benzoxazine you synthesized, bisphenol A, aniline and formaldehyde based, typically abbreviated as BA-a, have been published more than 100 times and they all show Tg close to 170oC which is one of the lowest among all benzoxazines studied. Did you study many benzoxazine papers published in the past? This reviewer was very surprised to find that this is the first paper of all the benzoxazine papers published in the past that does not cite even a single paper published by the benzoxazine research pioneer, Prof. H. Ishida. If you ever read many of his papers, you would never make such statements as he extensively reported high performance polybenzoxazines that show Tgs between 200-450oC. Even more shocking is that Prof. Xu is a long time experts on benzoxazines. If he read the manuscript carefully, this statement would not have passed his approval.
Page 4, line 154
State the proton, carbon and silicon frequencies used for the analysis.
Page 4, section 2.5
The name of the technique, unless you want to indicate the model names, should not be spelled in capital letters. Thus, “Fourier Transform Infrared Sopectrometry” should be “Fourier transform infrared spectrometry” Other names should be in the same way.
Page 5, line 175
The 1409 cm-1 band is the CH in-plane bending mode of the vinyl group on the silicon atom. Are you certain that the frequency of this band is 1409 cm-1. Could it be 1411 cm-1?
Page 7, line 219
The broad band around 3400 cm-1 is due to the OH group of the ring openened epoxy resin. The phenolic portion of the benzoxazine should be mostly around 3200-2500 due to the 6-membered intramolecular H-bond formed. A small amount of the phenolic OH group is intermolecularly hydrogen bonded and will should around 3300-3100 cm-1. This little contribution is seen at 3400 cm-1.
Page 8, Figure 9
Why 0.5% sample shows no evidence of crosslinking, reflecting from the even lower rubbery plateau around 250oC? Then, all of a sudden, it jumps with 1% sample. Is this result reproducible? The epoxy group on DDSQ-EP should copolymerize with the oxazine ring, especially because the DDSQ-EP at 0.5% should not be aggregated. Why also there is no property changes for 1, 2 and 4% samples. Again, are these results reproducible?
In conclusion, the experimental results were rather carefully done. However, it appears that the authors, at least the author who wrote the paper, are not familiar with the benzoxazine papers and some statements that are quite contrary to the reported facts are made. Upon minor correction of the problem stated above, the paper is recommended for acceptance in Polymers.
Author Response
Please see the PDF file in the attachment.

Author Response

(The authors gave the same response as above.)

Reviewer 3 Report
A very good report on new type of polybenzoxazine resins with improved properties. The experimental part is very well done and in good agreement with the conclusions. It is important to note that the field of polybenzoxazines remains a quite narrow one which brings few citations since the applications of the new materials are still to be developed at industrially scale.
Author Response

(The authors gave the same response as above.)

Reviewer 4 Report
The article is devoted to the actual topic. The description of the methods and the discussion need to be improved. My comments:
1. There is no information about the equipment on which the dielectric properties were studied.
2. Add dielectric study modes.
3. What shape were the samples for measuring the dielectric properties. How were they made?
Author Response

(The authors gave the same response as above.)

Round 2
Reviewer 2 Report
Accept in present form